# Community Integration of Persons with Mental Disorders Compared with the General Population

**DOI:** 10.3390/ijerph17051596

**Published:** 2020-03-02

**Authors:** Min Hwa Lee, Mi Kyung Seo

**Affiliations:** 1Department of Social Welfare, Mokpo National University, Muan 58554, Korea; hwasuri79@gmail.com; 2Department of Social Welfare, Gyeongsang National University, Jinju 52828, Korea

**Keywords:** physical integration, psychological integration, social integration, social network size, social contact frequency, community integration

## Abstract

*Aims*: Community integration is the catalyst for recovery that is provided by mental health services to persons with mental disorders. This study explores the impact of socio-demographic variables on the level of community integration in persons with mental disorders compared to the general population living in the same communities and the difference in community integration level between the two groups. *Methods*: A total of 224 persons with mental disorders (M *age* = 45.0, SD = 12.84, male 51.8%, female 48.2%) in communities and 247 individuals (M *age* = 44.6, SD = 11.41, male 50.6%, female 49.4%) of the general population in the same communities participated in the evaluation of levels of physical, psychological, and social integration. The effects of socio-demographic variables on the three types of community integration on both groups were evaluated using multiple regression analyses. Differences in the three types of community integration between the two groups were tested using multivariate analysis of covariance (MANCOVA) by controlling for socio-demographic variables as covariates. *Results*: The effects of socio-demographic variables on the three types of community integration differed between the two groups. In addition, the two groups differed significantly in terms of social rather than physical or psychological integration when the level of community integration was compared while controlling socio-demographic variables. The results also show that persons with mental disorders had smaller social networks and fewer social contacts than the general population. *Conclusions*: Based on the findings, we recommended that service providers provide incentives for consumers to strengthen social relationships and social skills training in order to maintain relationships.

## 1. Introduction

The aim of mental health serves is to recover persons with mental disorders. “Recovery” refers to persons with mental disorders experiencing “themselves as recovering a new sense of self and of purpose within and beyond the limits of the disability” [1]. In the process of recovery, persons with mental disorders pursue their life goals and maintain a satisfactory life in their communities. Recovery is, therefore, possible on the premise that their integration into local communities, as their context of life, must precede. Community integration is both a facilitator and an outcome of the recovery process [2]. It is also the most important predictor of quality of life in the community of persons with mental disorders [3,4,5,6].

Despite the importance of community integration, in most societies, persons with mental disorders are still marginalized. Their social networks are small and provide a low level of social support, and because of social stigma, they have limited opportunities for employment, housing, and education [7,8,9,10,11]. In Korea, their social exclusion is even more severe. Just 8.3% of all mentally ill people have jobs, 10.2% have housing insecurity, and only 2.4% use community services [12]. Mental health services are, therefore, faced with the tasks of actively supporting psychosocial aspects of persons with mental disorders to help them integrate into their community.

Originally the concept of community integration focused on physical integration (i.e., activities outside the home), but gradually evolved to include psychological and social integration. In their concept of community integration, Wong and Solomon [13] included the individual’s capacity to carry out daily activities in the community (physical integration), to pursue interaction with other members of their community without mental illness (social integration), and to feel a sense of belonging in their communities (psychological integration). Based on these concepts, physical integration is defined in this study as the degree to which an individual engages in activities and use resources outside their home. Social integration has two dimensions, a dimension of social network and the degree to which an individual interacts socially with others; thus, the size of social network and the frequency of the social contact are used. Psychological integration, in turn, defines the degree to which an individual feels they belong to the community.

Previous community integration studies have identified predictors of community integration of persons with mental disorders, including socio-demographic variables [14], psychopathology and social functions [15,16,17,18], service program characteristics [15,19,20], and neighborhood characteristics [18,21,22]. These studies were conducted on the premise that community integration of persons with mental disorders is generally low. Nevertheless, several studies have found that the level of social integration in persons with mental disorders is not lower compared to the general population and non-disabled persons [23,24] or that there are little, if any, differences [25]. On the other hand, some researches showed evidence that their level of community integration was clearly lower than others [2,26,27].

This study compared the level of community integration of the general population and consumers of mental health services with a common geographic and cultural characteristics in order to assess the extent to which the presence of mental disorders is a barrier to community integration. Based on the results, evidence-based intervention for community integration may be suggested. In addition, we explored how socio-demographic variables explain the differences in community integration between the two groups. In other words, the analyses focused on the extent to which socio-demographic variables account for community integration of both the general population and persons with mental disorders; physical, psychological, and social integration differs between the two groups when socio-demographic variables are controlled as covariates.

## 2. Methods

### 2.1. Participants

The study was approved from the Institutional Review Board of Gyeongsang National University. Participants included two samples: 247 general populations and 224 persons with mental disorders who were receiving community mental health services in Gyeongsangnam-do of Korea. After explaining the study specifics to the participants, we got written informed consent from the participants who volunteered to participate.

*General population:* We set the number of the general population over 20 by the age-appropriate participants in each region using area sampling and proportionate stratified sampling based on National Statistics Korea Population Statistics (2015), then recruited the participants who perform social activities without any mental health problems through a convenience sampling. The number of men and women were equally distributed. Among 247 adults, 125 (50.6%) were male and 122 (49.4%) were female. The mean age was 44.6 (SD = ±11.41); the mean educational years was 14.75 (SD = ±2.48); the average monthly income was 5,028,600 (SD = ±590.98) KRW. In terms of city size, 121 (49%) lived in large cities and 126 (51%) lived in small and medium-sized cities.

*Participants with mental disorders:* We recruited 224 participants with mental disorders over the age of 20 years from community mental health services facilities (mental hospitals, community mental health centers, and community rehabilitation centers) which cooperated with the study. The inclusion criteria for participants with mental disorders included a diagnosis of schizophrenia or a mood disorder according to DSM-5; living in the community and maintaining ongoing contact with the outpatient unit. Among them, 116 (51.8%) were male and 108 (48.2%) were female. The mean age was 45 (SD = ±12.84); the mean educational years was 12.36 (SD = ±3.17); the average monthly income was 1,679,000 (SD = ±162.5) KRW. Regarding the size of the city, 115 (51.3%) lived in large cities and 109 (48.7%) lived in small and medium-sized cities.

Chi-square tests were conducted to determine whether socio-demographic characteristics of the two groups were homogeneous. There were no statistically significant differences in gender ($\chi^{2}\;=\;$ 0.065, *p* = 0.798), age ($\chi^{2}$ = 49.467, *p* = 0.574), city size ($\chi^{2}$ = 0.260, *p* = 0.610), community resource accessibility ($\chi^{2}$ = 29.285, *p* = 0.740) between the general population and persons with mental disorders. However, the average monthly income ($\chi^{2}$ = 210.327, *p* < 0.001) and educational years ($\chi^{2}$ = 79.875, *p* < 0.001) were statistically significant differences between the two groups. In the general population, average monthly income and education levels were higher than in persons with mental disorders.

### 2.2. Measures

All participants were evaluated using a self-report based on socio-demographic and background variables including gender, age, educational years, average monthly income, and city size. Participants with mental disorders were given additional questions about their diagnosis.

*Physical integration:* To measure physical integration, we used the External Integration Scale [28]. Thirteen items assessed the individual’s frequency of involvement in different outdoor activities, such as eating in a restaurant, visiting a library, and walking in a park. Each item is rated from 1 (never) to 5 (very often), in which higher scores suggested higher levels of physical integration. Cronbach’s α for physical integration was 0.743.

*Social integration:* Social integration assessed the quantity of social relationships by social network size and social contact frequency. Social network size was measured by the number of families, relatives, friends, neighborhoods, or peers in touch over the past year. Items in the social contact frequency asked respondents how often they have had different types of social contact with family, relatives, friends, neighborhoods, or peers over the past year ranging from relatively superficial (such as saying hello) to closer contact (such as going out). Social contact frequency is scored from 1 (never) to 9 (almost every day), with higher scores indicating higher levels of social integration. Cronbach’s α for social integration was social network size 0.631; social contact frequency 0.594.

*Psychological integration:* In order for psychological integration, we used the Neighborhood Cohesion [29]. Ten items assessing the perceived sense of community belonging are rated from 1 (completely not true) to 5 (completely true), in which higher scores suggested higher levels of psychological integration. Cronbach’s α for psychological integration was 0.868.

### 2.3. Statistical Analysis

Statistical analyses were conducted with SPSS version 21.0 (SPSS Inc., Chicago, IL, USA). The descriptive statistical analysis was carried out to examine the socio-demographic characteristics of two groups, and we used chi-square tests to evaluate the homogeneity of the socio-demographic variables between the two groups. Regression analysis was used to analyze the socio-demographic variables’ effects on three types of community integration in both groups. Multivariate analysis of covariance (MANCOVA) was performed to investigate the differences in community integration scores between two groups. MANCOVA was calculated with the physical, social, and psychological community integration scores as dependence variables and groups (general population vs. persons with mental disorders) as independent variables and gender, age, educational years, average monthly income, and city size as covariates.

## 3. Results

### 3.1. Effect of Socio-Demographic Variables on Community Integration

The effects of socio-demographic variables on three types of community integration among two groups were analyzed using regression analysis (Table 1). In terms of physical integration, educational years (β = 0.16) and city size (β = −0.20) were statistically significant predictors for persons with mental disorders. For those living in large cities and higher educational levels, the level of physical integration was higher. On the other hand, socio-demographic variables did not significantly affect the level of physical integration of the general population. With regard to social integration and social network size, the first type of social integration, has been significantly affected by all variables except gender in persons with mental disorders. With age (β = 0.19), years of education (β = 0.19), monthly income (β = 0.21), and living in a small- or medium-sized city (β = 0.15), social network size increased. In comparison, the social network size in the general population was significantly affected only by age (β = 0.31). Gender (β = 0.18) and years of education (β = −0.20) were the significant predictors of the general population’s social contact frequency—the second type of social integration. Social contact frequency among women was higher and the educational level was lower; the frequency of social contact was higher. In comparison, no social-demographic variable significantly affected social contact frequency in persons with mental disorders. Finally, age was the only significant predictor of psychological integration for both groups. In fact, the level of psychological integration in both the general population (β = 0.34) and those with mental disorders (β = 0.25) increased with age. In the general population, city size (β = 0.17) also predicted psychological integration, with higher psychological integration levels occurring in small and medium-sized cities.

### 3.2. Comparison between Persons with Mental Disorders and General Populations

We compared means for the physical, social, and psychological community integration between the general population and persons with mental disorders controlling gender, age, educational years, average monthly income, and city size. As can be seen in Table 2, there was a significant group difference in the social integration, but there was no significant difference in physical integration (F = 0.003, df = 1; 424, *p* = 0.958) and psychological integration (F = 0.237, df = 1; 424, *p* = 0.627) between the two groups, based on Wilks’ criterion (F = 30.45, df = 4; 416, *p* < 0.001).

Social network size (F = 104.64, df = 1; 424, *p* < 0.001), the first type of social integration, showed significant differences between the two groups. For the general population, social network size was greater (3.80 ± 0.71) than for those with mental disorders (2.39 ± 0.98). It suggests there are more family members, relatives, friends, neighbors, or peers in the general population than do persons with mental disorders. For social network size, four of the five covariates, age (t = 4.93, *p* < 0.001, power = 0.998), educational years (t = 2.45, *p* < 0.05, power = 0.687), monthly income (t = 3.78, *p* < 0.001, power = 0.965), and city size (t = 2.50, *p* < 0.05, power = 0.704) are significantly related but gender is not. In terms of covariate power for social network size, age was followed by monthly income, educational years, and city size.

Social contact frequency (F = 30.01, df = 1; 424, *p* < 0.001), the second type of social integration, showed significant differences between the two groups. For the general population, social contact frequency was also higher (6.16 ± 1.23) than for persons with mental disorders (4.98 ± 1.64). The results suggest that the general population has more frequent contacts with family members, relatives, friends, neighbors, or peers than do persons with mental disorders. None of the five covariates are related to social contact frequency.

## 4. Discussion

Several notable findings have emerged from the current study. First, the predictor variables of the community integration differed between persons with mental disorders and the general population based on the results of the regression analysis conducted to analyze the impact of socio-demographic variables on community integration. Gender was, above all, a significant predictor of social contact frequency in this study for the general population only. Nevertheless, previous studies indicated that either for persons with mental disorders or for the general population, gender was not significantly associated with community integration [16,24,25,30,31,32].

Age had a significant effect on social network size and psychological integration in both groups in this study. In fact, older people had larger social networks and higher levels of psychological integration. It is possible that, when a person ages, the social network will expand; the level of psychological integration will also increase depending on the length of time they live in the community. Nevertheless, research findings on the relationship between age and community integration are inconsistent. While many studies reported no associations between age and community integration of persons with mental disorders [15,16,30,32], others reported evidence of their relationship. Such results have shown that as age increases, physical integration decreases [33], and overall community integration decreases [31]; psychological integration increases on the other hands [34]. As far as the general population is concerned, a study examining all three types of community integration found no significant relationship between them and age [25], while a study examining only psychological integration reported evidence of their association [24].

In the present study, educational level had a significant effect on the frequency of social contact among the general population only; however, it had a positive effect on physical integration and social network size in persons with a mental disorder. Monthly income also had a significant effect on social network size among persons with mental disorders alone. In this study, the educational level and monthly income of both the general population (r = 0.254) and persons with mental disorders (r = 0.182) were significantly correlated. The two variables are important indicators of socioeconomic status. The results also indicate that, as socioeconomic status increases, social network size may increase. This is consistent with the study by Yanos et al. [25] on a significant relationship between educational levels and social integration and the findings by Abdallah et al. [2] that community integration is growing as income increases. Most of the studies, however, argued that at a significant level, the educational level does not predict community integration [15,24,31]. In terms of city size, the physical integration of persons with mental disorders in large cities was higher than in small and medium-sized cities, while the general population’s psychological integration in small and medium-sized cities was higher than in large cities. It contradicts the results of Kruzich [33] that small cities have a greater positive impact on the physical integration of persons with mental disorders than large cities. Just a few studies focused on the relationship between city size and community integration; it is, therefore, difficult to conclude. In general, physical integration in large cities is likely to be higher because they usually have more resources available, while their sense of community is likely to be higher in small cities due to less anonymity, although the results are inconsistent between the two groups.

The second important finding of the present study is that when socio-demographic variables were controlled as covariates, the two groups differed in terms of social integration rather than physical or psychological integration. In other words, compared to the general population, persons with mental disorders have smaller social networks and fewer social contacts. This result is consistent with Aubry and Myner’s finding [23] that used similar methods to compare the two groups. Although there was no significant difference in psychological integration, there was an apparent difference in social integration—but only in comparing the general population with persons with mental disorders when controlling for socio-demographic variables. In a study that examined psychological integration alone, there was no significant difference between persons with mental illness and non-disabled community members [24]. Research focusing on social networks, a type of social integration, found that persons with mental disorders scored lower in both quantitative and qualitative dimensions of the social network [35]. Unlike the present study, another study that compared mental health consumers living in supported housing with neighborhood residents showed the former a slightly lower level of overall community integration [25]. Additionally, some studies reported significantly lower levels of community integration for persons with mental disorders compared to the general population [2,26]. Nevertheless, these studies have not been able to control variables that may influence community integration; thus, it is impossible to determine whether the differences are due to mental disorder or other variables.

There are several implications for the significant difference in social integration between persons with mental disorders and the general population found in this study. First, a small social network for persons with mental disorders who experience their disorders becoming chronic with lower social functioning means that they are socially isolated and faced challenges in obtaining the social support needed to live in their communities. Such isolation poses the risk that their psychopathology will deteriorate. Second, despite people with mental health problems having a level of physical integration similar to that of the general population, a lower level of social integration means that the activities of persons with mental disorders do not contribute to building meaningful social relationships. Furthermore, even if there is a “sense of community” among those with mental disorders at a level similar to that of the general population, the presence of a low level of social integration also suggests that a sense of community still does not provide opportunities for social relationships. Third, social integration is the most important predictor of quality of life among the three areas of community integration [34,35,36]. Low social integration is, therefore, seen as a challenge to overcome for their quality of life and recovery.

## 5. Conclusions

A comparison of persons with mental disorders and the general population living in the same communities has yielded significant findings. On the basis of the results, the authors propose the following. First, mental health service providers must develop programs aimed at encouraging consumers of mental health services to build relationships with other community members and provide them with opportunities to actively engage in community activities. Second, although there is a need to broaden their social networks, it is important to help them develop close relationships with frequent contacts. Such intimate relationships are the source of social support and are essential to building long-term stable relationships. Intervention is therefore needed to help them maintain relationships not only with professionals and other persons with mental disorders, but also with close friends and family members who can understand and support them in the community, i.e., taking recreational activities, self-help group programs, contacts with family, or psychotherapy groups. Thirdly, before expanding their social networks, it is also important to improve their potential for social relationships such as social skills, and the ability to use and access social media.

There are, however, several limitations. First, the assessment was limited to quantitative areas of social integration (i.e., social network size and contact frequency) and did not include qualitative aspects. Since individuals can maintain close relationships in small networks, in which the quality rather than the quantity of relationships can have a greater significance in terms of social support, future research needs to take careful consideration qualitative aspects. Second, the size of the city was the only characteristic of the community included in this study and is insufficient to represent the cultural characteristics of the community. Future research, therefore, needs to include additional variables to assess the characteristics of the community (e.g., tolerance for the acceptance of people with mental disorders, stigma and crime rate against them) in order to inform community-specific practical interventions.

## Figures and Tables

**Table 1 ijerph-17-01596-t001:** Effect of socio-demographic variables on community integration.

	General Population (*n* = 238, Social Network Size *n* = 236)			Persons with Mental Disorders (*n* = 192, Social Network Size *n* = 190)		
	*β*	t	*p*	*β*	t	*p*
**Physical Integration** (1)						
Gender	0.06	0.94	0.350	0.06	0.81	0.421
Age	0.09	1.34	0.182	0.02	0.30	0.766
Years of education	0.07	1.03	0.302	0.16	2.18	0.030
Average monthly income	0.11	1.63	0.104	0.08	1.08	0.282
City size	0.02	0.30	0.766	−0.20	−2.80	0.006
**Social Integration**						
**Social Network Size** (2)						
Gender	−0.06	−0.89	0.372	0.06	0.78	0.436
Age	0.31	4.79	<0.001	0.19	2.51	0.013
Years of education	0.07	0.96	0.336	0.19	2.63	0.009
Average monthly income	0.11	1.74	0.083	0.21	2.77	0.006
City size	0.09	1.47	0.143	0.15	2.17	0.031
**Social Contact Frequency** (3)						
Gender	0.18	2.87	0.004	0.03	0.38	0.708
Age	−0.04	−0.53	0.598	0.06	0.76	0.447
Years of education	−0.20	−2.95	0.003	0.14	1.85	0.065
Average monthly income	0.08	1.22	0.224	0.08	0.95	0.346
City size	0.04	0.56	0.574	−0.001	−0.01	0.990
**Psychological Integration** (4)						
Gender	0.07	1.16	0.249	0.04	0.56	0.575
Age	0.34	5.34	<0.001	0.25	3.09	0.002
Years of education	−0.01	−0.22	0.829	0.13	1.69	0.092
Average monthly income	0.10	1.51	0.132	0.08	1.03	0.306
City size	0.17	2.86	0.005	−0.02	−0.33	0.744

Notes: a. General population, b. participants with mental disorders. (1) a. F = 1.402, df = 5 and 232, *p* = 0.224;$\;R^{2}=\;$0.029, b. F = 3.302, df = 5 and 186, *p* = 0.007;$\;R^{2}=\;$0.082. (2) a. F = 6.334, df = 5 and 230, *p* < 0.001; $R^{2}=\;$0.121, b. F = 4.645, df = 5 and 184, *p* = 0.001; $R^{2}$ = 0.112. (3) a. F = 3.510, df = 5 and 232, *p* = 0.004; $R^{2}$ = 0.070, b. F = 0.990, df = 5 and 186, *p* = 0.425; $R^{2}$ = 0.026. (4) a. F = 8.770, df = 5 and 232, *p* < 0.001; $R^{2}$ = 0.159, b. F = 2.256, df = 5 and 186, *p* = 0.051; $R^{2}$ = 0.057. Gender (1 = male, 2 = female), city size (1 = large cities, 2 = small or medium cities).

**Table 2 ijerph-17-01596-t002:** Comparison of community integration between groups by MANCOVA.

	General Population (*n* = 236)	Persons with Mental Disorders (*n* = 190)	MANCOVA (1)
	M (SD)	M (SD)	
Physical integration	3.58 (0.51)	3.44 (0.66)	F = 0.003, df = 1; 424, *p* = 0.958
Social integration			
Social network size	3.80 (0.71)	2.39 (0.98)	F = 104.64, df = 1; 424, *p* < 0.001
Social contact frequency	6.16 (1.23)	4.98 (1.64)	F = 30.01, df = 1; 424, *p* < 0.001
Psychological integration	3.39 (0.70)	3.19 (0.73)	F = 0.237, df = 1; 424, *p* = 0.627

(1) Gender, age, years of education, average monthly income, and city size as covariates.

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
