# Peer review of "Community Integration of Persons with Mental Disorders Compared with the General Population"

_ijerph, 2020, doi:10.3390/ijerph17051596_

Round 1
Reviewer 1 Report
The study entitled “Community Integration of Persons with Mental Disorders Compared with the General Population” explore the impact of socio-demographic variables on the level of community integration in persons with mental disorders compared to the general population living in the same communities and the difference in community integration level between the two groups.
The paper is well written and organized, but I don’t recommend the publication of the present version. I highlight minor issues that should be addressed before the manuscript can be considered for publication.
Material and Methods
It is recommendable to include in the procedure of data collection that an Ethic Committee approved the study and talk about the inform consent (I suppose that it was like that) The criterial participants with mental disorders are very clear. But for general population more information should be mentioned. For example, how do you know that the general population does not present any mental disorder? It is recommendable to include more information about the statistical analysis. The explanation about the statistical analyses carried out is not sufficiently clear.Conclusion
Line 272, the following sentence “This section is not mandatory, but may be added if there are patents resulting from the work reported in this manuscript.” should be deleted.References
Please, review the references. There are some mistakes.Author Response
Thanks for your comments.
"Please see the attachment."
We replied to point in blue with your comment.
1) Line 79, 81-83
It included in the procedure of data collection.
2) Line 86-88
The general participants were recruited by convenience sampling, including colleagues, friends, neighbors, mates, and members of the fraternity etc. And they perform social activities without any mental health problems. So, it included the content.
3) Line 131-140
It included additional information on the statistical analysis.
4) Line 272 --- revised Line 280 : Delite
5) Line 300-400
We were revising the reference.

Reviewer 2 Report
I found the paper interesting, well written and clear. The originality is on average, but it could take new point of view on this filed giving interesting operative suggestions:
I intend that it could be interesting too add some clinical and practical suggestion to ameliorate the social support perceptions both in the families and between peers, i.e. adopting recreational activities or contacts with familiaris or psychotherapy groups so that patients could implement their social contacts.
This comment could be add in the discussion or in the conclusion sections.
Even though the paper is good, and the English is OK. I also have some other comments for this paper:
In the tables it suggest to put in the numbers only two decimals after the comma(Table 1 and Table 2) and it could be indicated the p value with 0.001(not 0.000). The references are pertinent addressing important link to the discussed topic and formally corrected.I have not concerns about the paper. Compliments, a good work.
Author Response
Thanks for your comments.
"Please see the attachment."
We replied to point in red with your comment.
1) table 1 and table 2
We revised the numbers on two decimals after the comma
and indicated p value with 0.001.
2) Line 267-269
Some suggestion for your comment was added.
3) Line 300-400
We were revising the reference.
